# High Perceived Susceptibility to and Severity of COVID-19 in Smokers Are Associated with Quitting-Related Behaviors

**DOI:** 10.3390/ijerph182010894

**Published:** 2021-10-17

**Authors:** Yajie Li, Tzu Tsun Luk, Yongda Wu, Derek Yee Tak Cheung, William Ho Cheung Li, Henry Sau Chai Tong, Vienna Wai Yin Lai, Sai Yin Ho, Tai Hing Lam, Man Ping Wang

**Affiliations:** 1School of Nursing, The University of Hong Kong, Hong Kong, China; yajieli@connect.hku.hk (Y.L.); yongdang@connect.hku.hk (Y.W.); derekcheung@hku.hk (D.Y.T.C.); mpwang@hku.hk (M.P.W.); 2Nethersole School of Nursing, Chinese University of Hong Kong, Hong Kong, China; william3@hku.hk; 3Hong Kong Council on Smoking and Health, Hong Kong, China; lawrence_chu@cosh.org.hk (H.S.C.T.); ed@cosh.org.hk (V.W.Y.L.); 4School of Public Health, The University of Hong Kong, Hong Kong, China; syho@hku.hk (S.Y.H.); hrmrlth@hku.hk (T.H.L.)

**Keywords:** risk perception, quit attempt, smoking cessation, tobacco, coronavirus disease, Chinese

## Abstract

A growing body of evidence shows smoking is a risk factor for coronavirus disease (COVID-19). We examined the associations of quitting-related behaviors with perceived susceptibility to and severity of COVID-19 in smokers. We conducted a telephone survey of 659 community-based adult smokers (81.7% male) in Hong Kong, where there was no lockdown. Exposure variables were perceptions that smoking can increase the risk of contracting COVID-19 (perceived susceptibility) and its severity if infected (perceived severity). Outcome variables were quit attempts, smoking reduction since the outbreak of the pandemic, and intention to quit within 30 days. Covariates included sex, age, education, heaviness of smoking, psychological distress, and perceived danger of COVID-19. High perceived susceptibility and severity were reported by 23.9% and 41.7% of participants, respectively. High perceived susceptibility was associated with quit attempts (prevalence ratio (PR) 2.22, 95% CI 1.41–3.49), smoking reduction (PR 1.75, 95% CI 1.21–2.51), and intention to quit (PR 2.31, 95% CI 1.40–3.84). Perceived severity of COVID-19 was associated with quit attempts (PR 1.64, 95% CI 1.01–2.67) but not with smoking reduction or intention to quit. To conclude, the perceived susceptibility to and severity of COVID-19 in smokers were associated with quitting-related behaviors in current smokers, which may have important implications for smoking cessation amid the pandemic.

## 1. Introduction

Non-pharmacological measures taken to contain the coronavirus disease 2019 (COVID-19) pandemic, such as social distancing and lockdowns, have had far-reaching impacts on activities of daily life, including health-related behaviors. Studies have found both increased and reduced levels of smoking and motivation to quit among cigarette smokers since the COVID-19 outbreak [1,2,3]. Greater psychological distress was found to be associated with both increased and reduced levels of smoking in cross-sectional studies [4,5], and financial instability was found to predict smoking cessation [6]. Other factors associated with changes in smoking behaviors amid the pandemic have remained understudied, especially in places where lockdowns have not been implemented.

The World Health Organization warned that smoking may increase the risk of infection with and severity of COVID-19 [7]. Systematic reviews have consistently shown that smoking is a risk factor for COVID-19 complication and death [8,9,10,11]. In contrast, the link between smoking and COVID-19 infection is less clear. Tobacco smoke exposure is a known cause of acute respiratory tract diseases (e.g., pneumonia) and impaired immunity [12]. The hand-to-mouth routine of smoking and mask removal may predispose smokers to virus exposure from contaminated hands and cigarettes, as well as from people around [13]. However, a lower rate of COVID-19 infection among smokers than among never smokers has also been reported [14,15,16]. These findings are difficult to interpret because of methodological problems, such as collider bias [17], misclassification of smoking status, and reverse causality (stopping smoking because of COVID-19 symptoms). Two population-representative surveys in the UK found higher odds of self-reported confirmed COVID-19 infection among current versus never smokers [18,19]. A study of 2.4 million users of a COVID-19 symptom app also suggested that smokers had a higher risk of developing symptomatic COVID-19 [20]. Yet, these studies were limited by their cross-sectional design and the reliance on self-reported data.

Studies that have suggested nicotine may reduce COVID-19 infection [21] have often been mistaken as definitive evidence supporting the protective role of smoking and have had negative impacts. Our population-based survey found that one fifth of the participants had seen claims via social media that smoking can protect against COVID-19, and such exposure was associated with increased levels of smoking in smokers [5]. People’s perceptions may underlie the impact of (mis)information exposure on behavioral changes [22] and thus may have implications for public health campaigns aiming to promote smoking cessation amid the pandemic. Hong Kong, the most Westernized city in China, has the lowest daily smoking prevalence (10.2% in 2019) in the developed world, but the decline in smoking here has become stagnant in recent years [23]. This study aims to examine the associations of quitting-related behaviors since the outbreak of the pandemic with perceived susceptibility to and severity of COVID-19 due to smoking in Hong Kong current smokers.

## 2. Materials and Methods

### 2.1. Study Design

A telephone survey was conducted from May 7 to June 30, 2020, which was between the peaks of the 2nd (March) and 3rd (August) waves of the COVID-19 outbreak in Hong Kong, with sporadic local cases appearing. The Hong Kong government had enforced social distancing measures, such as limiting gatherings to not more than 4 people, but no lockdown had been implemented. The Institutional Review Board of the University of Hong Kong/Hospital Authority Hong Kong West Cluster approved the survey (UW 20-326).

### 2.2. Sampling Methods

We collected data from former participants in a territory-wide, community-based smoking cessation contest conducted in 2018 (ClinicalTrials.gov, number NCT03565796) and 2019 (NCT03992742). This contest has been organized yearly since 2009 by the Hong Kong Council on Smoking and Health for promoting smoking cessation using competition and prizes. Daily-smoking adults in the community were recruited from all 18 districts in Hong Kong and given a brief behavioral cessation intervention. All contestants verified their smoking status with an exhaled carbon monoxide level of 4 parts per million or higher and were followed for 6 months. Those who successfully quit during the contest with biochemical validation were awarded up to HK$1000 (≈US$125). The main results of the contest in 2018 are reported elsewhere [24].

In May to July 2020, as a separate study, we re-contacted the contestants for a telephone survey that aimed to examine the impacts of COVID-19 on smoking. To be eligible for the survey, participants needed to be Hong Kong residents aged 18 years or older, to currently smoke, and to able to communicate in Chinese. Since the study focused on quitting behaviors during the COVID-19 outbreak, those who successfully quit during the contest (i.e., before the outbreak) were excluded. We successfully re-contacted 968 potential participants, and 769 (79.4%) agreed to participate in the present survey. After excluding people who reported having quit smoking (*N* = 80) and those who were not aware of COVID-19 (*N* = 30), 659 current smokers remained for analysis. Participants who were not aware of COVID-19 were mostly (68.8%) people over 60 years old.

### 2.3. Data Collection

All the participants provided verbal informed consent during the telephone interview, which took 10 to 15 min to complete.

The perceived susceptibility of smokers to COVID-19 was assessed by asking “to what extend do you agree that smoking can increase the risk of contracting COVID-19?”; perceived severity was assessed by asking “to what extend do you agree that smokers are more likely than non-smokers to have severe consequences, such as need of intensive care and mechanical ventilation and deaths, after contracting COVID-19?”. Responses to both questions were dichotomized into “Yes (Strongly agree/Agree)” and “No (Neutral/Disagree/Strongly disagree)”.

To assess quit attempts, the participants were asked “have you ever made a serious quit attempt since the outbreak of COVID-19?” (Yes/No). Participants also reported whether they had changed their amount of smoking since the outbreak, with responses of “Increased”, “No change”, and “Reduced” recorded. A few participants reported having increased their consumption (*N* = 39) and were excluded when smoking reduction was analyzed as the outcome. The stage of change measure was adapted to measure intention to quit [25], in which the participants were asked “when are you going to quit smoking?” with 4 response options. This was analyzed as whether the participant intended to quit within 30 days, which was classified as “No” (undecided/within 6 months) or “Yes” (within 30 days/within 7 days).

We also collected data on potential confounders, including heaviness of smoking [26], psychological distress [4,5], and perceived danger of COVID-19. The heaviness of smoking index ranged from 0 to 6, with a higher score indicating greater smoking dependence [27]. Psychological distress was measured by the Patient Health Questionnaire-4 (PHQ-4, range 0–12), with a score of 6 or above indicating the presence of depressive and anxiety symptoms [28]. The perceived danger of COVID-19 was assessed on a scale of 0 (not dangerous at all) to 10 (very dangerous). Data on sex, age, and education were also collected.

### 2.4. Statistical Analysis

All statistical analyses were performed using Stata/MP version 15.1 (StataCorp LLC, College Station, TX, USA). Chi-Square tests were used to compare the 3 types of quitting-related behaviors (quit attempt, smoking reduction, and intention to quit) according to different perceptions of smoking, COVID-19, and characteristics. We used Poisson regressions with robust variance estimators to estimate the prevalence ratio (PR) for each quitting-related behavior in relation to the perception of smoking and COVID-19 [29], adjusting for sociodemographic factors, heaviness of smoking, psychological distress (PHQ-4), and perceived danger of COVID-19. For common binary outcomes (>10% in the sample), PR can be interpreted as relative risk and thus more advantageous to odds ratio estimated by logistic regressions. As a sensitivity analysis, we calculated the e-value for each observed association, which is the strength (in RR scale) of an unmeasured confounder required to attenuate the observed association to null [30]. A higher e-value suggests that a stronger unmeasured confounder is needed to nullify the observed association. All statistical tests were 2-sided, with *P* < 0.05 indicating statistical significance.

## 3. Results

### 3.1. Participant’s Characteristics

Table 1 shows that, of the 659 participants, 87.1% were male, 45.2% were aged 40 to 59 years, 39.0% had senior secondary education, 43.3% had a moderate to high level of smoking, and 13.5% had psychological distress (PHQ-4 score ≥ 6). Our sample was largely comparable with the census data on smokers in Hong Kong in terms of sex (87.1% vs. 83.4%), age (45.2% vs. 47.3% aged 40–59 years), and mean cigarette consumption (12.1 vs. 12.7 cigarettes per day) [23].

### 3.2. Perceived Susceptibility and Severity

Table 1 shows that the overall prevalence of a high perceived susceptibility to and severity of COVID-19 in relation to smoking was 23.9% (95% CI 20.5–27.5%) and 41.7% (37.6–45.9%), respectively. Sociodemographic characteristics, heaviness of smoking, and psychological distress were similar (*P* > 0.05), but the prevalence of perceived severity was lower in those with senior secondary education (*P* = 0.034).

### 3.3. Quitting-Related Behaviors 

Table 2 shows that 17.9% (95% CI 15.1–21.1%) had attempted to quit smoking, 27.9% (23.0–29.7%) had reduced smoking consumption and 14.3% (11.8–17.2%) intended to quit in 30 days. Perceived susceptibility was significantly associated with higher prevalence of all 3 quitting-related behaviors (all *P* < 0.001). Participants with perceived severity had higher prevalence of quit attempt (*P* < 0.001) and intention to quit in 30 days (*P* = 0.01) than those without, but the prevalence of smoking reduction was similar (*P* = 0.12). Moderate/high (vs low) heaviness of smoking was negatively associated with quit attempt (*P* = 0.001) and reduced smoking (*P* < 0.001). 

Table 3 shows that perceived susceptibility was associated with quit attempts (PR 2.22, 95% CI 1.41–3.49%; *P* < 0.001; e-value 3.87), smoking reduction (PR 1.75, 95% CI 1.21–2.51%; *P* = 0.003; e-value = 2.90), and intention to quit in 30 days (PR 2.31, 95% CI 1.40–3.84%; *P* = 0.001; e-value = 4.05) after adjustment. Higher perceived severity (PR 1.64, 95% CI 1.01–2.67%; *P* = 0.046; e-value = 2.66) and higher perceived danger of COVID-19 (PR 1.13, 95 % CI 1.02–1.25%; *P* = 0.015; e-value = 1.51) was associated with quit attempts.

## 4. Discussion

We found that, amid the COVID-19 pandemic, current smokers’ perceived susceptibility to COVID-19 due to smoking was associated with a 75% to 130% increased likelihood of engaging in quitting-related behaviors (smoking reduction, quit attempts, and intention to quit within 30 days), while perceived severity of COVID-19 due to smoking was associated with a 64% increased likelihood of engaging in quit attempts. These behaviors are well-established antecedents of successful quitting [26,31]. Our findings suggested that the ongoing COVID-19 pandemic could be an opportune moment in which to encourage smoking cessation.

We examined two dimensions of risk perceptions, susceptibility and severity, which are key constructs in models of behavioral change, such as the Health Belief Model [32] and the Health Action Process Approach model [33]. Previous studies have found that a high perceived risk of developing a smoking-related disease was predictive of motivation to quit, quit attempts, and successful quitting [34,35]. A recent experimental study found that smokers exposed to messages regarding the link between smoking and COVID-19 showed a higher perceived effectiveness of attempts to discourage smoking than those who were unexposed [36]. These findings, together with ours, suggest that heightening smokers’ perceptions of their susceptibility to and the severity of COVID-19 in relation to smoking could encourage smoking cessation in the context of the pandemic; thus, randomized controlled trials focusing on providing information to enhance smokers’ knowledge of COVID-19 and heighten their perception of the severity of COVID-19 are warranted.

We found that fewer participants perceived a high susceptibility compared to those who perceived a high severity (23.9% vs. 41.7%). This might reflect the stronger evidence supporting the association of smoking with COVID-19 progression [7,8,9,10,11] than that on COVID-19 infection at the time of our data collection, which is still the case. Exposure to unverified but widely disseminated claims that smoking can prevent COVID-19, as reported in another study [37] and ours [5], might also explain this discrepancy. The spread of this misinformation should be curbed and smokers’ misperceptions of smoking and COVID-19 should be corrected urgently. Importantly, public health messaging should be based on information with substantiated evidence. While further studies are needed to clarify the association between nicotine intake and COVID-19 infection, the greater risk of severe illness seen in current smokers infected with COVID-19 should be emphasized to encourage smoking cessation. The potential risk of infection due to unmasking and gathering at smoking hotpots (public areas where ashtrays are available) could also be highlighted, especially in places where no lockdown has been implemented.

The COVID-19 pandemic has presented an unprecedented context in which to study novel COVID-19-related factors relating to changes in smoking and quitting behaviors. Emerging studies have mostly been conducted in places where lockdowns have been implemented, causing substantial disruptions in daily life activities and thus smoking behaviors [1,2,3,4,6]. Our study was conducted in a place where no lockdown was enforced, precluding its potential impact on smoking behaviors. Our findings could still be explained by other factors caused by the pandemic, such as financial instability [6], home isolation, and unknown factors that have not been reported. Therefore, in addition to the adjustment of known confounders, we calculated the e-values to examine the sensitivity of our findings to unmeasured confounders. The large e-values obtained of 1.51 to 4.05 mean that an unmeasured confounder needs to be associated with both perceptions and quitting-related factors with an RR of 1.51 to 4.05 to nullify the corresponding associations. As this seems unlikely, the e-values support the validity of our findings.

Our study had some limitations. First, the study was cross-sectional and causation could not be confirmed. However, it seems less likely that quitting-related behaviors could influence people’s perception of smoking and COVID-19. Prospective studies are warranted to confirm these findings. Second, as in other related studies [1,2,3,4,5,19,38,39], our data were self-reported, which might have made them subject to social desirability bias. Third, the participants only reported whether they had increased smoking, not changed their smoking behaviors or decreased smoking since the pandemic. This may have led to measurement error, because it is possible that some smokers both increased and decreased their level of smoking during the pandemic [39]. Prospective studies with repeated measures of smoking consumptions are needed in order to understand the complex relations between COVID-19-related factors and smoking behaviors. Fourth, unmeasured confounders cannot be excluded in observational studies, but the large e-values found suggest that the results were unlikely to have been substantially affected by confounders. We also adjusted for heaviness of smoking, a well-established predictor of smoking cessation [26], among other factors. Finally, excluding those who had quit after the onset of COVID-19 might lead to an underestimation of the associations if quitting was due to perceived susceptibility or perceived severity. However, excluding this group made the sample more homogeneous, as all were current smokers and therefore able to report their intention to quit.

## 5. Conclusions

This study found that perceived COVID-19 susceptibility and severity in relation to smoking were significantly associated with quit attempt, smoking reduction, and intention to quit.

## Figures and Tables

**Table 1 ijerph-18-10894-t001:** Prevalence of high perceived susceptibility to and severity of COVID-19 in relation to smoking according to participants’ characteristics.

Characteristics	Total (*N* = 659)	Perceived Susceptibility ^a^ (*N* = 137)		Perceived Severity ^b^ (*N* = 228)	
	*N* (%)	% ^c^	*P* ^d^	% ^c^	*P* ^d^
Overall		23.9		41.7	
Sex			0.75		0.86
Male	574 (87.1)	23.6		41.5	
Female	85 (12.9)	25.3		42.7	
Age, years			0.075		0.15
18–39	246 (39.0)	23.2		41.7	
40–59	285 (45.2)	21.7		38.3	
60+	99 (15.7)	34.2		51.5	
Education			0.065		0.034
Junior secondary or below	204 (34.2)	28.7		48.5	
Senior secondary	233 (39.0)	19.6		36.2	
Tertiary	160 (26.8)	28.3		46.5	
Heaviness of smoking ^e^			0.192		0.65
Low	330 (56.7)	26.2		40.9	
Moderate/High	282 (43.3)	21.2		42.9	
Psychological distress ^f^			0.87		0.63
<6	526 (86.5)	23.8		40.7	
≥6	82 (13.5)	24.7		43.7	

^a^ Agreed that smoking can increase the risk of COVID-19 infection. ^b^ Agreed that smoking can increase the risk of COVID-19 progression. ^c^ Row percentage. ^d^ P value was calculated by Chi-square test. ^e^ Scores range from 0 to 6; scores were rated as low (0–2), moderate (3–4), or high (5–6). ^f^ Assessed by the Patient Health Questionnaire 4 (scores range 0–12), with a score of 6 or above indicating psychological distress.

**Table 2 ijerph-18-10894-t002:** Prevalence of high perceived susceptibility and severity of COVID-19 in relation to smoking according to participants’ characteristics.

Characteristics	Quit Attempt (*N* = 116)		Smoking Reduction (*N* = 172)		Intention to Quit in 30 Days (*N* = 94)	
	% ^a^	*P* ^b^	% ^a^	*P* ^b^	% ^a^	*P* ^b^
Overall	17.9		27.9		14.3	
Perceived susceptibility ^c^		<0.001		<0.001		<0.001
No	12.1		21.6		10.8	
Yes	37.0		44.3		25.6	
Perceived severity ^d^		<0.001		0.12		0.013
No	10.0		24.7		11.2	
Yes	28.5		30.8		18.6	
Sex		0.83		0.64		0.48
Male	18.0		28.2		14.6	
Female	17.1		25.6		11.8	
Age, years		0.72		0.09		0.36
18–39	19.5		31.7		13.4	
40–59	16.8		23.2		16.5	
60+	17.7		30.1		11.1	
Education		0.21		0.99		0.49
Junior secondary or below	23.0		28.0		15.2	
Senior secondary	17.0		28.0		13.7	
Tertiary	16.9		28.6		18.1	
Heaviness of smoking ^e^		0.001		<0.001		0.11
Low	22.5		34.2		17.6	
Moderate/High	11.6		19.4		12.7	
Psychological distress ^f^		0.56		0.17		0.74
<6	17.1		28.6		14.5	
≥6	19.8		20.8		15.9	

^a^ Row percentage. ^b^ P value was calculated by Chi-square test. ^c^ Agreed that smoking can increase the risk of COVID-19 infection. ^d^ Agreed that smoking can increase the risk of COVID-19 progression. ^e^ Scores range 0–6; scores were rated as low (0–2), moderate (3–4), or high (5–6). ^f^ Assessed by the Patient Health Questionnaire 4 (scores range 0–12), with a score of 6 or above indicating psychological distress.

**Table 3 ijerph-18-10894-t003:** Association of perceptions of smoking and COVID-19 with quitting-related behaviors.

	Quit Attempt		Smoking Reduction		Intention to Quit in 30 Days	
	PR (95% CI) ^a^	*P*	PR (95% CI) ^a^	*P*	PR (95% CI) ^a^	*P*
Perceived susceptibility ^b^						
No	1		1		1	
Yes	2.22 (1.41–3.49)	<0.001	1.75 (1.21–2.51)	0.003	2.31 (1.40–3.84)	0.001
Perceived severity ^c^						
No	1		1		1	
Yes	1.64 (1.01–2.67)	0.046	1.01 (0.71–1.44)	0.96	1.02 (0.61–1.71)	0.94
Perceived danger of COVID-19 ^d^	1.13 (1.02–1.25)	0.015	1.03 (0.95–1.11)	0.51	1.10 (0.97–1.24)	0.14

^a^ Calculated by Poisson regression model with robust variance, adjusting for sex, age, education, heaviness of smoking, psychological distress (PHQ-4), and other variables listed in the table. ^b^ Agreed that smoking can increase the risk of COVID-19 infection. ^c^ Agreed that smoking can increase the risk of COVID-19 progression. ^d^ Scores range 0 to 10, with higher scores indicating greater perceived danger of COVID-19; PR is for per score increased.

## Data Availability

The data presented in this study are available on reasonable request from the corresponding author. The data are not publicly available, because the data will only be available upon the approval of the funder.

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
