# Peer review of "High Perceived Susceptibility to and Severity of COVID-19 in Smokers Are Associated with Quitting-Related Behaviors"

_ijerph, 2021, doi:10.3390/ijerph182010894_

Round 1
Reviewer 1 Report
This study sets out to ask whether there is a relationship between smokers beliefs as to whether smoking affects their susceptibility to or severity of COVID-19, and their smoking/quitting behaviour. The study appears to be well thought out and conducted, and well and clearly written up.
The results are neither contentious nor surprising, but are useful. We should not assume that something is so just because it makes sense. I found it interesting that the idea of being more susceptible to getting COVID-19 had more effect than the fear that if a smoker got it they would have a worse outcome. That's a pity, since the evidence for the latter is so much more consistent.
I found myself really wanting to know about the four whose smoking increased, but maybe there's nothing obvious by way of reason for that.
The authors conclusions that these beliefs appear to affect quitting behaviour seem justified and are approached with due consideration of potential ways in which they could be misleading. Their suggestion that fear of COVID-19 could create a "teachable moment" to encourage cessation is reasonable.
I found only one typographical error. In the abstract, line 17, "variable" should be "variables" .
My major problem with this paper comes about because my understanding of the relationship between smoking and susceptibility to COVID-19 is that it smoking has a real possibility of being partially protective and that, if it is, this is likely to be because of the effects of nicotine. The authors rightly acknowledge that it is contentious, but I think an expanded view of the literature on this would be useful (e.g. a very recent paper by Lee and co-workers https://doi.org/10.1093/ntr/ntab079. There are others taking a similar view. I disagree that information in such papers is "misinformation", although it is true that COVID-19 misinformation abounds).
Leaving the impression that the controversy is settled in favour of smoking adding to susceptibility needs to be better established, if the authors wish to maintain that. While the papers cited are reputable, they are not without interpretational difficulties themselves. For example, collider bias (ref 17) could work in the opposite way to that proposed in a country where asymptomatic COVID infections are common, as in the UK. Further, the study (ref 20) may well have been strongly biased by smokers being more likely to report coughing.
The authors meant to cite a comment on the paper by Changeaux (ref 21) not the Changeaux paper itself. I note that this comment does not dispute that nicotine could be protective, but dislikes any implication that smoking is protective.
I understand that this matter is contentious and also that no-one wants to encourage smokers to keep on smoking. Furthermore, COVID is changing. It seems that younger people are becoming more susceptible. Why should susceptibility of smokers not vary between different variants? But perhaps something could be said in the discussion about any "teaching" needing to be based on factual information, and that the clearest message that can be given is that smokers are much more likely to become seriously ill if they do catch COVID-19.
Author Response
C1: This study sets out to ask whether there is a relationship between smokers beliefs as to whether smoking affects their susceptibility to or severity of COVID-19, and their smoking/quitting behaviour. The study appears to be well thought out and conducted, and well and clearly written up.
The results are neither contentious nor surprising, but are useful. We should not assume that something is so just because it makes sense. I found it interesting that the idea of being more susceptible to getting COVID-19 had more effect than the fear that if a smoker got it they would have a worse outcome. That's a pity, since the evidence for the latter is so much more consistent.
I found myself really wanting to know about the four whose smoking increased, but maybe there's nothing obvious by way of reason for that. The authors conclusions that these beliefs appear to affect quitting behaviour seem justified and are approached with due consideration of potential ways in which they could be misleading. Their suggestion that fear of COVID-19 could create a "teachable moment" to encourage cessation is reasonable.
R1: We would like to thank the Reviewer for the detailed comments and the helpful suggestions to strengthen the paper, especially on how to provide a more balanced view of the literature on the link between smoking, nicotine, and susceptibility to COVID-19. We have now avoided framing smoking as a definite risk factor in the introduction and discussion. Please refer to our responses to specific comments below for details.
C2: I found only one typographical error. In the abstract, line 17, "variable" should be "variables".
R2: Thanks for noting the error. We have revised as “variables” in line 17.
C3: My major problem with this paper comes about because my understanding of the relationship between smoking and susceptibility to COVID-19 is that it smoking has a real possibility of being partially protective and that, if it is, this is likely to be because of the effects of nicotine. The authors rightly acknowledge that it is contentious, but I think an expanded view of the literature on this would be useful (e.g. a very recent paper by Lee and co-workers https://doi.org/10.1093/ntr/ntab079. There are others taking a similar view. I disagree that information in such papers is "misinformation", although it is true that COVID-19 misinformation abounds).
R3: We agree that an expanded view of the literature is needed and thank the reviewer for the useful paper, which has now been cited as followed (ref 16): “However, a lower rate of COVID-19 infection among smokers than never-smokers was reported [14-16]”.
Furthermore, we have now avoided the word “misinformation” throughout the article when referring to studies that suggested a protective role of nicotine in COVID-19 infection. For instance, in line 61-63: “Our population-based survey found that one-fifth of the participants had seen misinformation claims via social media that smoking can protect against COVID-19…”
C4: Leaving the impression that the controversy is settled in favour of smoking adding to susceptibility needs to be better established, if the authors wish to maintain that. While the papers cited are reputable, they are not without interpretational difficulties themselves. For example, collider bias (ref 17) could work in the opposite way to that proposed in a country where asymptomatic COVID infections are common, as in the UK. Further, the study (ref 20) may well have been strongly biased by smokers being more likely to report coughing.
R4: Thanks very much for pointing out the need of a more balanced discussion on the link between smoking and COVID-19 susceptibility, which we very much agree. Apart from the suggested changes as responded in R3 above, we have now tuned down our wordings “The findings may have been misinterpreted were difficult to interpret because of methodological problems, such as collider bias [17], misclassification of smoking status, and reverse causality (stopping smoking because of COVID-19 symptoms). We have now also noted the limitations of studies that suggested a higher risk of COVID-19 in current smokers as followed: “Yet, these studies were limited by the cross-sectional design and the reliance on self-reported data.”
C5: The authors meant to cite a comment on the paper by Changeaux (ref 21) not the Changeaux paper itself. I note that this comment does not dispute that nicotine could be protective, but dislikes any implication that smoking is protective.
R5: We have now revised the statement as followed “Studies that suggested nicotine may reduce COVID-19 infection [21] had often been mistaken as definitive evidence to support a protective role of smoking with negative impact.”
C6: I understand that this matter is contentious and also that no-one wants to encourage smokers to keep on smoking. Furthermore, COVID is changing. It seems that younger people are becoming more susceptible. Why should susceptibility of smokers not vary between different variants? But perhaps something could be said in the discussion about any "teaching" needing to be based on factual information, and that the clearest message that can be given is that smokers are much more likely to become seriously ill if they do catch COVID-19.
R6: We agree that COVID is changing and that the risk of COVID-19 may also differ among variants of the virus, although investigating the associations between smoking and COVID-19 susceptibility was not the focus of our study. As suggested, we have now added in the discussion that “Importantly, public health messaging should be based on information with substantiated evidence. While further studies are required to clarify the association between nicotine intake and COVID-19 infection, the greater risk of severe illness in current smokers infected by COVID-19 and smoking behaviors could be emphasized to encourage cessation. The potential risk of infection due to unmasking and gathering at smoking hotspots could also be highlighted, especially in places where there was no lockdown.”
Reviewer 2 Report
The study "Perceived susceptibility to and severity of COVID-19 from smoking were associated with quitting-related behaviors" aimed to "examine the associations of quitting-related behaviors with perceived susceptibility to and severity of COVID-19 from smoking".
- First of all this study requires moderate English changes (especially style).
"Drastic measures" is not scientific wording.
"Growing evidence shows smoking is associated with coronavirus disease (COVID-19)." - this sentence is misleading - smoking was perceived as a risk factor for COVID-19 but not directly associated with COVID-19.
Similar sentences should be cleared through the text. - "Unsubstantiated claims that smoking can protect against COVID-19 have been widely circulated online with negative impacts [21]. " - most of the studies suggested (wrongly) that nicotine can protect against COVID-19, not smoking.
- The Authors should clearly define the aim of the study.
- The methods described in this study are inadequate to address the study's aim. Please provide the logical structure of the methods. Sampling should be clearly defined. Tools used to assess smoking status should be precisely described. "Details of the methods of the contest have been reported elsewhere [24]."- this sentence is unacceptable. Methods are crucial to understand this paper and evaluate the scientific soundness of this study.
- Most of the participants were male (87%). This may lead to a serious risk of bias and the results can not be generalized to the whole population.
- Line 150 - doubled headline
- Analyses presented in Table 3 are very simple. Please provide a more advanced model.
- Conclusions should be based on the own findings of the authors. The current conclusions are weak and do not refer to the findings obtained by the Authors.
Author Response
C1: First of all this study requires moderate English changes (especially style).
"Drastic measures" is not scientific wording.
R1: We would like to thank the reviewer for the comments and suggestions for improving our paper. We have now removed “drastic” and revised the sentence into “The non-pharmacological measures to contain the Coronavirus 2019 (COVID-19) pandemic, such as social distancing and lockdown, have far-reaching impacts on daily life activities including health behaviors.” in line 32-34.
C2: "Growing evidence shows smoking is associated with coronavirus disease (COVID-19)." - this sentence is misleading - smoking was perceived as a risk factor for COVID-19 but not directly associated with COVID-19.
Similar sentences should be cleared through the text.
R2: Thanks for the comment. As suggested, we have now revised the sentence in line 14 as “Growing evidence shows smoking is a risk factor for coronavirus disease (COVID-19).”.
Similarly, in the main text: “Systematic reviews have consistently shown that smoking was associated with a risk factor for COVID-19 complications and deaths [8-11].”
C3: "Unsubstantiated claims that smoking can protect against COVID-19 have been widely circulated online with negative impacts [21]." most of the studies suggested (wrongly) that nicotine can protect against COVID-19, not smoking.
R3: Thanks for noting the nuance between smoking, nicotine and COVID-19 infection, which was also pointed out by Reviewer 1 (see C5 above). We have now revised the sentence as followed “Studies that suggested nicotine may reduce COVID-19 infection [21] had often been mistaken as definitive evidence to support a protective role of smoking with negative impacts.”
C4: The Authors should clearly define the aim of the study.
R4: We have now clearly stated the aim of our study at the end of the introduction (line 68-70) as “This study aims to examine the associations of quitting-related behaviors since the pandemic with perceived susceptibility to and severity of COVID-19 from smoking in Hong Kong current smokers.”
C5: The methods described in this study are inadequate to address the study's aim. Please provide the logical structure of the methods. Sampling should be clearly defined. Tools used to assess smoking status should be precisely described. "Details of the methods of the contest have been reported elsewhere [24]."- this sentence is unacceptable. Methods are crucial to understand this paper and evaluate the scientific soundness of this study.
R5: We agree that further information on the sampling needed to be included and have now restructured the Methods and added “2.2 Sampling methods”. Details of sampling, including method to assess smoking status, has been further elaborated as follow: “We collected data from former participants in a territory-wide, community-based smoking cessation contests conducted in 2018 (ClinicalTrials.gov, number NCT03565796) and 2019 (NCT03992742). Details of the methods of the contest have been reported elsewhere [24]. The contest has been organized yearly since 2009 by the Hong Kong Council on Smoking and Health for promoting smoking cessation by competition and prizes. Daily smoking adults were recruited in the community from all 18 districts in Hong Kong and given brief behavioral cessation intervention. All contestants verified their smoking status with an exhaled carbon monoxide level of 4 parts per million or higher and were followed for 6 months. Those who successfully quit during the contest with biochemical validation were awarded up to HK$1000 (≈US$125). The main results of the contest in 2018 were reported elsewhere [24].
In May to July 2020, as a separate study, we re-contacted the contestant for a telephone survey which aims to examine the impacts of COVID-19 on smoking. To be eligible for the present survey, participants needed to be Hong Kong residents aged 18 years or older, currently smoke, and able to communicate in Chinese. Since the study focused on quitting behaviors during the COVID-19 outbreak, those who successfully quit during the contest (i.e., before the outbreak) were excluded. We successfully re-contacted 968 potential participants, and 769 (79.4%) agreed to participate in the present survey. After excluding people who reported having quit smoking (n=80) and those who were not aware of COVID-19 (n=30), 659 current smokers remained for analysis. Participants who were not aware of COVID-19 were mostly (68.8%) people over 60 years old.”
C6: Most of the participants were male (87%). This may lead to a serious risk of bias and the results can not be generalized to the whole population.
R6: Thanks for the comment. As have been mentioned in the results (line 149-152), “our sample was largely comparable with the census data on smokers in Hong Kong in sex (87.1% vs 83.4%), age (45.2% vs 47.3% aged 40–59 years) and mean cigarette consumption (12.1 vs 12.7 cigarettes per day)”. The results that most of the participants were male was consistent with the male predominance of smoking in the Chinese and Asian cultures. We believe our results could be generalized to the underlying population.
C7: Line 150 - doubled headline
R7: Thanks for noting the error. We have deleted the doubled headline in line163.
C8: Analyses presented in Table 3 are very simple. Please provide a more advanced model.
R8: We have considered different statistical models for binary outcomes to analyze the associations reported in Table 3. Since the prevalence of all 3 quitting-related outcomes were relatively prevalent (>14%), the odds ratio (OR) of outcomes calculated by binary logistic regression would overestimate the effect size when interpreted as risk ratio (RR). Therefore, we have opted for Poisson regression with robust variance to estimate the prevalence ratio (PR), which can be interpreted as RR for common binary outcome (Barros & Hirakata, 2003). We believe the modified Poisson regression model is appropriate given the data available in our study. Furthermore, as mentioned in the footnote of the Table 3, all analyses were adjusted for potential confounding factors. We have now provided further information about the regression model in the footnote of Table 3 as followed: “a Calculated by Poisson regression model with robust variance, adjusting for sex, age, education, heaviness of smoking, psychological distress (PHQ-4), and other variables in the table were adjusted.”
Reference: Barros, A.J., Hirakata, V.N. Alternatives for logistic regression in cross-sectional studies: an empirical comparison of models that directly estimate the prevalence ratio. BMC Med Res Methodol 3, 21 (2003). https://doi.org/10.1186/1471-2288-3-21
C9: Conclusions should be based on the own findings of the authors. The current conclusions are weak and do not refer to the findings obtained by the Authors.
R9: We have now ensured the conclusion only refer to our own findings as followed “This study found that perceived COVID-19 susceptibility and severity in relation to smoking were significantly associated with quit attempt, smoking reduction, and intention to quit.”
Round 2
Reviewer 1 Report
The Authors provide some revisions but responses to the comments are not very detailed. There are still major concerns about the methodology. Moreover, the link between smoking and COVID-19 analyzed in this study was not explain sufficiently.
Moreover, the novelty of these findings is limited. This paper may be submitted to some local journal, due to the limited novelty that would be interesting for the international readers.
Author Response
Thank you very much for your further comments on our paper. We apologize if our responses fell short of addressing your concerns and would be very grateful if you could specify which part of the methodology requires additional information. Moreover, we would like to clarify that our study aims to examine the link between smoking-related perceptions of COVID-19 and quitting-related behaviours, not the link between smoking and COVID-19.
As discussed in the paper, we believe our study uniquely add to the literature by showing the factors associated with quitting-related behaviors under the COVID-19 pandemic in a place without lockdown. This is a strength since lockdown could strongly influence the changes in health behaviors and confound the associations between other factors associated with cessation under the pandemic.